# Variation in Survival and Gut Microbiome Composition of Hatchery-Grown Native Oysters at Various Locations within the Puget Sound

Emily Kunselman,[a] Jeremiah J. Minich,[b] Micah Horwith,[c] Jack A. Gilbert,[a,d,e] Eric E. Allen[a,e,f]

aCenter for Marine Biotechnology and Biomedicine, Scripps Institution of Oceanography, University of California, San Diego, La Jolla, California, USA
bThe Plant Molecular and Cellular Biology Laboratory, The Salk Institute for Biological Studies, La Jolla, California, USA
cWashington State Department of Ecology, Seattle, Washington, USA
dDepartment of Pediatrics, School of Medicine, University of California, San Diego, La Jolla, California, USA
eCenter for Microbiome Innovation, University of California, San Diego, La Jolla, California, USA
fMolecular Biology Section, Division of Biological Sciences, University of California, San Diego, La Jolla, California, USA

**ABSTRACT** The Olympia oyster (*Ostrea lurida*) of the Puget Sound suffered a dramatic population crash, but restoration efforts hope to revive this native species. One overlooked variable in the process of assessing ecosystem health is association of bacteria with marine organisms and the environments they occupy. Oyster microbiomes are known to differ significantly between species, tissue type, and the habitat in which they are found. The goals of this study were to determine the impact of field site and habitat on the oyster microbiome and to identify core oyster-associated bacteria in the Puget Sound. Olympia oysters from one parental family were deployed at four sites in the Puget Sound both inside and outside of eelgrass (*Zostera marina*) beds. Using 16S rRNA gene amplicon sequencing of the oyster gut, shell, and surrounding seawater and sediment, we demonstrate that gut-associated bacteria are distinct from the surrounding environment and vary by field site. Furthermore, regional differences in the gut microbiota are associated with the survival rates of oysters at each site after 2 months of field exposure. However, habitat type had no influence on microbiome diversity. Further work is needed to identify the specific bacterial dynamics that are associated with oyster physiology and survival rates.

**IMPORTANCE** This is the first exploration of the microbial colonizers of the Olympia oyster, a native oyster species to the West Coast, which is a focus of restoration efforts. The patterns of differential microbial colonization by location reveal microscale characteristics of potential restoration sites which are not typically considered. These microbial dynamics can provide a more holistic perspective on the factors that may influence oyster performance.

**KEYWORDS** gut microbiome, marine microbiome, oysters

Invertebrate microbiology research is increasingly important in the face of environmental and anthropogenic change. Olympia oyster (*Ostrea lurida*) populations declined across their native range on the west coast of the United States due to overharvesting by humans in the late 1900s (1). The loss of the Olympia oysters poses a threat to ecosystem services, as oysters create structured habitat and filter surrounding water (1). Recovery of these valuable services could be achieved through restoration efforts. To improve restoration outcomes, it is essential to identify where juvenile oysters will survive and grow successfully. Environmental and host-associated microbiota can impact settlement and growth in marine invertebrates (2–4) but the impact of microbial communities on the survival and growth of Olympia oysters in particular is unknown. Here, we explore the connection

Address correspondence to Emily Kunselman, ekunselm@ucsd.edu.

The authors declare no conflict of interest.

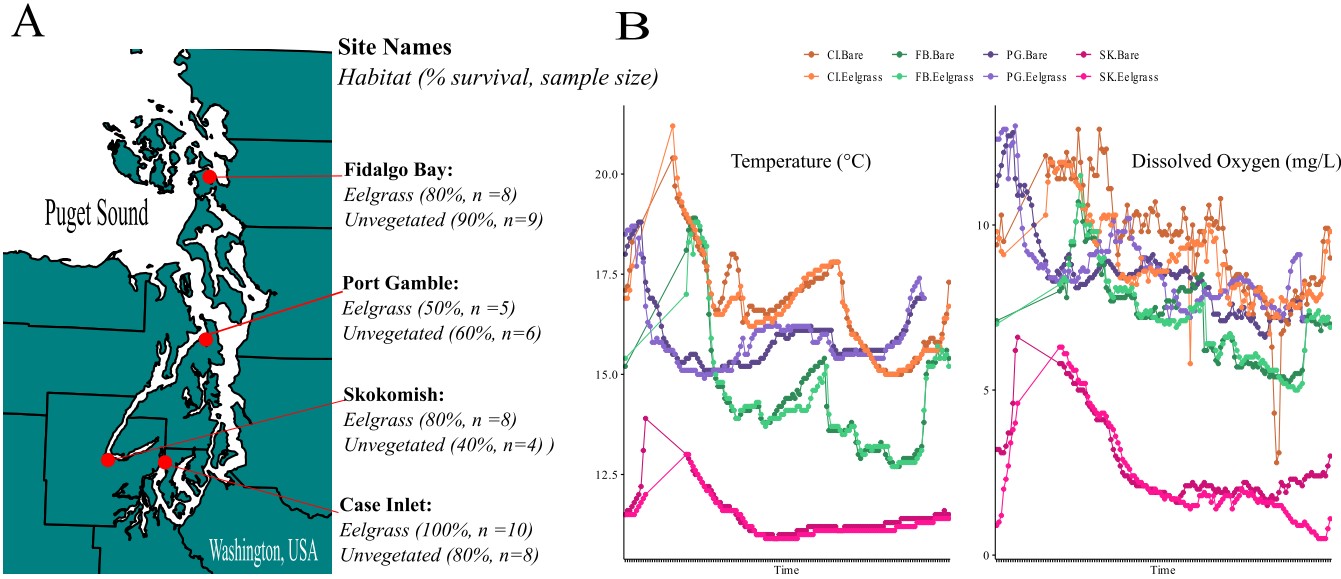

**FIG 1** Overview of study site characteristics. (A) Juvenile oyster survival rates across four field sites and two types of habitats in Washington State, USA (20 oysters initially deployed at each site). (B) Temperature and dissolved oxygen measurements at each site, for both habitats, plotted over the 24-h period prior to sampling.

between Olympia oyster performance and associated microbiota through a field experiment in Puget Sound, Washington (USA).

Temperature, dissolved oxygen, salinity, and carbonate chemistry can limit oyster growth, metabolism, and survival (5–7) and may therefore limit restoration success. Alone or in tandem, environmental conditions can stress oysters and make them more susceptible to disease (5, 8). Environmental stress can alter diversity and composition of oyster microbiomes, either as a result of bacterial response to the changing environment, or to the host's changing gene expression (9–11). A core microbiota has been demonstrated for oysters (12–14), but microbiota also vary significantly depending on environmental conditions and on the geographic location of the host (9, 10, 12, 13, 15, 16). A disturbance of the oyster microbiome may have consequences for host health due to the direct and indirect benefits of bacteria. Metabolism and enzyme production by bacteria in the gut improves digestion and provides additional nutrients to the host (17). Studies have shown that bacteria may prime the immune system and protect against pathogens (18). Some bacteria may have the ability to regulate oxidative stress through production of antioxidants (19). Finally, the oyster microbiome can indirectly benefit the host through production of antimicrobial peptides, which may limit growth of pathogens (20).

In this study, we evaluated the microbial diversity associated with the native Olympia oyster by comparing environmental and host-associated microbiota to identify differences across field sites and habitats and connections with oyster performance. The study aimed to: (i) characterize core or consistent members of the Olympia oyster microbiome, independent of other factors; and (ii) assess the extent of microbial variation across space. Methodologically, oysters were outplanted from a hatchery to field sites either inside or outside of eelgrass beds, left in place for 2 months, and then dissected and processed for bacterial community analysis. The field sites and habitats were further characterized by physicochemical parameters and assessment of the environmental microbiome.

## RESULTS

Oyster survival was highest at Case Inlet and Fidalgo Bay and lowest at Skokomish and Port Gamble (Fig. 1A). Difference in survival is significant between Port Gamble and Case Inlet (proportion test, $P = 0.0336$). Mean survival in eelgrass beds across all sites (mean = 77.5 ± 20.6%) was slightly higher than that of unvegetated habitat

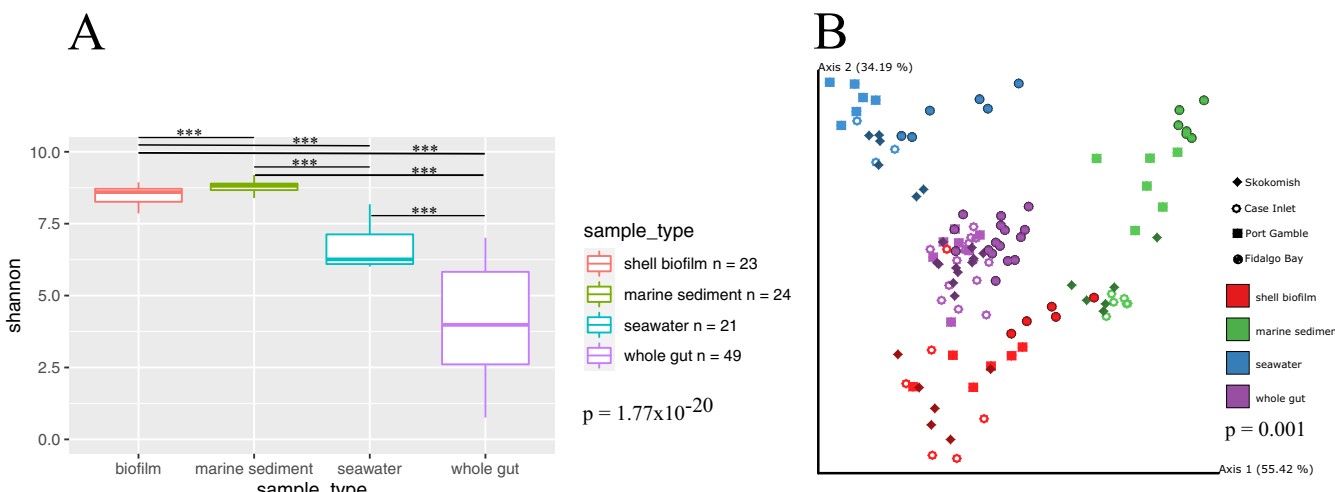

**FIG 2** Alpha (A) and beta (B) diversity across sample types: seawater and sediment (*n* = 3 per cage), oyster gut (*n* varies by cage due to differences in survival) and shell biofilm (*n* = 3 per cage). (A) Shannon Diversity Index used to calculate alpha diversity by sample type. Significance of pairwise comparisons is indicated by ***, which implies adjusted *P* < 0.001. (B) Robust Aitchison Principal Components Analysis plot demonstrating distance between sample types. Within each sample type grouping, different shapes are used to differentiate which study site the sample comes from. The RPCA metric was used to calculate the dissimilarity matrix and define top explanato*r*y axes.

(mean = 67.5 ± 22.2%), but this trend is not consistent across all sites and is not significant (two-way *t* test, *P* = 0.533). Alpha and beta diversity analyses were conducted on habitat type (eelgrass habitat versus unvegetated habitat) with considerations for nestedness. For alpha diversity, an ANOVA was run on habitat type and showed no interactions with geographic location or sample type (Data Set A2), although assumptions of normal distribution were violated to test this effect. For beta diversity, adonis was run on habitat type, which was nested within each site and across sample types (Data Set B2). Overall, no significant differences in alpha diversity or beta diversity among all samples were observed between habitats (Shannon ANOVA, F = 0.002, *P* = 0.962; Unweighted UniFrac Adonis, F = 1.257, *P* = 0.123). For this reason, habitat type was not considered for subsequent analyses.

Temperature was significantly different across the sites but did not vary between eelgrass and unvegetated habitat (PERMANOVA by site, F = 411.478, *P* = 0.0002, PERMANOVA by habitat, F = 0.33596, *P* = 0.5626, Data Set D2). Dissolved oxygen also varied significantly across site but not between habitats (PERMANOVA by site, F = 258.9586, *P* = 0.0002, PERMANOVA by habitat, F = 0.9197, *P* = 0.3266, Data Set D1). There were no interactions between site and habitat when comparing temperature or dissolved oxygen. These data were plotted by site and habitat and the Skokomish site showed the lowest values overall for both temperature and dissolved oxygen (Fig. 1B).

Alpha diversity (Shannon's index) was significantly different (Kruskal-Wallis, H = 95.084, *P* = 1.77 × 10$^{-20}$) between sample types (Fig. 2A, Data Set A1). All pairwise comparisons between groups were significant, indicating that sediment samples host the highest diversity, followed by biofilm, seawater, and oyster gut (Fig. 2A, Data Set A1). While oyster gut samples were found to host the lowest diversity of bacteria, they also manifest the greatest range in alpha diversity, suggesting that some samples were higher in richness and evenness than others (Fig. 2A). Robust aitchison principal component analysis (RPCA) analysis of beta diversity concluded that sample types varied significantly from one another in composition PERMANOVA, F = 123.43, *P* = 0.001; Fig. 2B, (Data Set B1). Pairwise comparisons in beta diversity between sample types show that they all are significantly different in composition (Data Set B1). As a reference, gut samples were closest in similarity to the biofilm samples ($\bar{x}_{distance}$ = 1.67, *P* < 0.001), followed by sediment samples ($\bar{x}_{distance}$ = 2.13, *P* < 0.001), and furthest in distance from seawater samples ($\bar{x}_{distance}$ = 2.28, *P* < 0.001).

Taxonomic alignment of bacteria amplicon sequence variants (ASVs) reveals relative abundances of key taxa groups within each sample type (Fig. 3A). Taxonomic

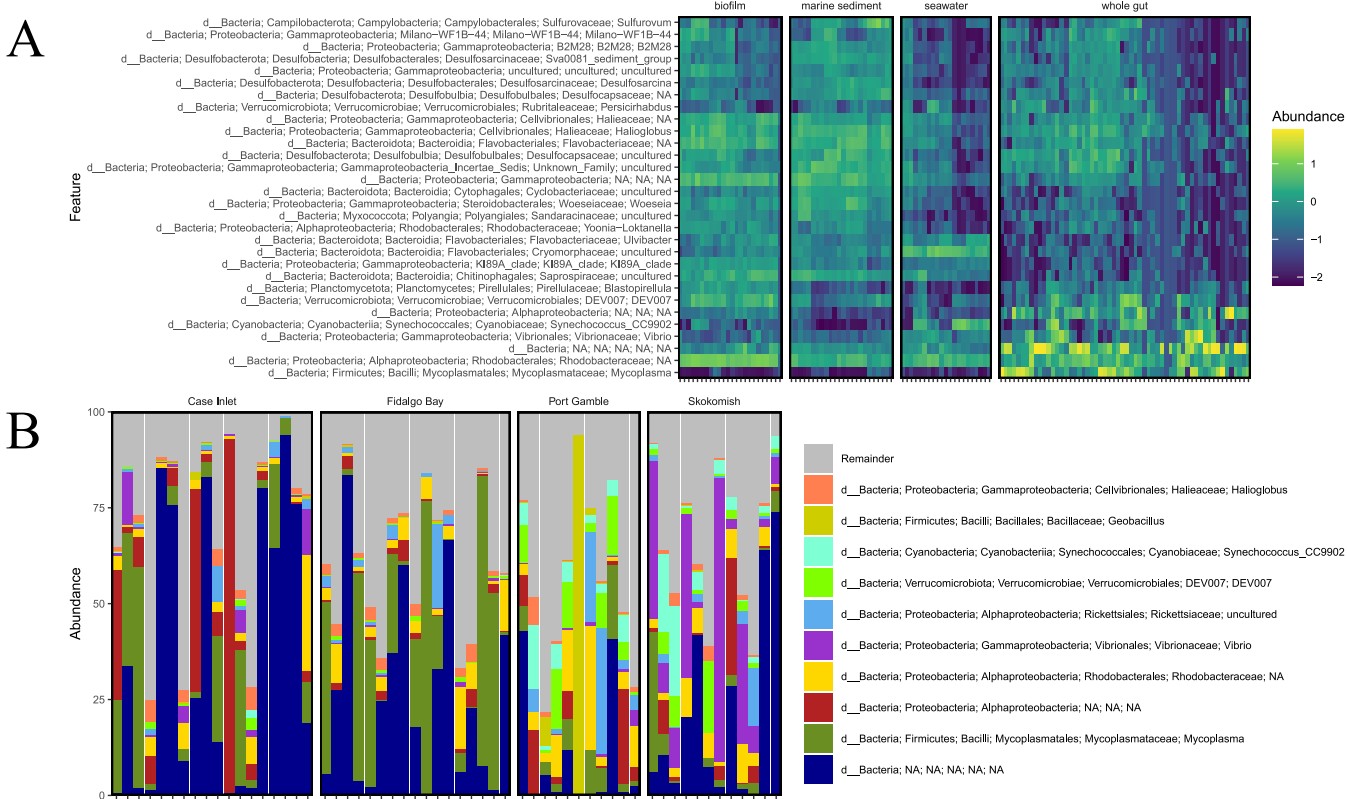

**FIG 3** Taxonomic composition assigned by comparison to the Silva database to identify bacterial groups across sample types. (A) Heat map comparing relative abundances of taxa across sample types. The scale assigns a positive number to taxa which comprise a large majority of their sample composition while negative numbers are assigned to taxa which comprise a minority of the sample or are completely absent. Abundances are not absolute, but rather the relative percentage unique to each sample showing patterns in the over or under representation of key taxa. (B) Taxa bar plot displaying relative abundances of major bacterial groups within oyster gut samples. The bar plot is separated by study sites after finding significant differences in the beta diversity of gut samples between different sites.

assignment of ASVs identified across samples demonstrates that *Mycoplasma* sp. dominates the oyster gut samples compared to any other sample type, which mostly lack *Mycoplasma* spp. (Fig. 3A). A large proportion of gut samples contain an unidentified ASV in relatively high abundance. This ASV was blasted against the NCBI 16S rRNA gene database to assess the nature of this sequence. The ASV was found to be only 87% similar to the closest match, which is *Nitrosomonas marina*. When placed in a phylogenetic tree, the ASV falls within a large group of Proteobacteria. This ASV was not filtered out of the data set during mitochondrial and chloroplast sequence exclusion and insertion tree placement, therefore, it is unlikely to be a eukaryotic sequence.

Although alpha diversity within the oyster gut samples did not significantly vary across sites (Kruskal-Wallis, H = 5.01, $P$ = 0.17, Data Set E), beta diversity did significantly vary across sites (PERMANOVA on RPCA distance matrix, F = 10.6534, $P$ = 0.001; Fig. 4, Data Set B3). DEICODE biplots were used to identify ASVs driving differences across sites. One ASV (pink *Vibrio* arrow, Fig. 4) appeared to drive separation of the Skokomish samples. This ASV was inspected closer by searching related sequences in the NCBI nucleotide blast database. Multiple species were 100% similar to this sequence, but the top hits were *Vibrio toranzoniae*, *Vibrio crassostreae*, and *Vibrio kanaloae*. Using Qurro, a visualization tool for the differentials generated by Songbird (21), ratios of the driving taxa were generated for the boxplot in Fig. 4 and values were organized by site. DEICODE and Songbird differentials can both be viewed in Qurro, but Songbird models are trained on metadata variables of interest and therefore the predictive accuracy of the model is directly related to the metadata variables included in the model's formula. The Songbird model that was generated with a formula of field site outperformed the null model with a $Q^2$ score, which is similar in concept to an $R^2$ value, of 0.17. For the ratio, groups of ASVs assigned to *Vibrio*, *Synechococcus*, and

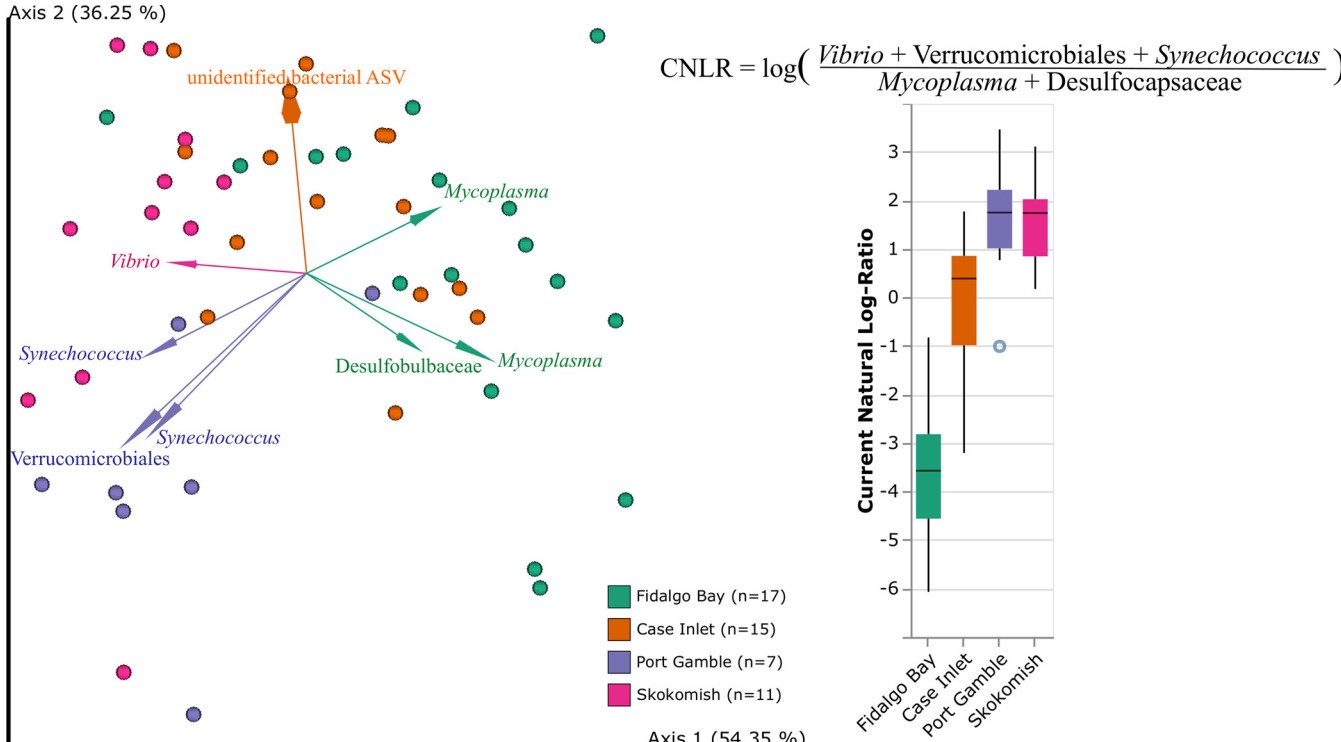

FIG 4 Variance in the oyster gut microbiota between sites. (Left) RPCA plot with only oyster gut samples. The dots are color coded by geographic location (site) within the Puget Sound and the arrows are colored by groups of bacterial ASVs found across samples which drive separation of that site. The RPCA biplot displays arrows which demonstrate the top 8 features associated with dissimilarity between samples. The visual association of these arrows with specific study sites informed the taxonomic groups to use for the differential abundance analysis ratios displayed in the box plot on the right. (Right) Ratio of differential abundances generated by Songbird analysis with *Vibrio*, Verrucomicrobiales, and *Synechococcus* aligned ASVs as the numerator and *Mycoplasma* and Desulfocapsaceae aligned ASVs as the denominator.

Verrucomicrobiales were clustered because they were heavily associated with oysters in the Port Gamble and Skokomish sites. ASVs assigned to *Mycoplasma* and Desulfocapsaceae were clustered because they appeared to drive the separation of the Fidalgo Bay oyster samples from other gut samples. Seven samples were dropped from the Qurro visualization because zeros were present in the log-ratio, implying that these ASVs were not actually detected in those samples. The comparison of relative abundances of a single taxon across samples can be misleading because its value within each sample depends on the abundance of all other taxa within that sample. To avoid this issue, one taxon is chosen as a reference and differentials of the other taxa are compared to this reference. This allows inference of the taxa's true change in relative abundance from one site to the next. *Mycoplasma* spp. were chosen as the reference because this group is found in the majority of gut samples, allowing for a consistent comparison of other groups from one site to the next. The "identify core features" command was used to identify ASVs present in over 75% of gut samples, one of which was a *Mycoplasma* ASV. After the differentials of specific taxa are grouped into the ratio, with the reference group in the denominator, the natural log is taken, and these values are plotted in Fig. 4.

*Mycoplasma* and Desulfocapsaceae ASVs were at greater proportions in the samples at Fidalgo Bay than *Vibrio*, Verrucomicrobiales, and *Synechococcus* ASVs. Port Gamble and Skokomish demonstrated the opposite trend: *Vibrio*, Verrucomicrobiales, and *Synechococcus* ASVs were at a greater proportion than *Mycoplasma* and Desulfocapsaceae ASVs. Case Inlet represents a middle ground, where the ratio fluctuates around 0 to show that these specific ASVs were overall fairly equal in abundance for the group of samples from this site. While this ratio does not come from absolute abundances and therefore, we cannot define the midpoint of the x axis, the use of reference points from the differential abundance analysis confirms the observation that these taxa explain variation between sites. The natural log ratio values were imported into R and run through a Kruskal Wallis nonparametric analysis of variance test and

found to be significantly different across sites (H = 33.243, $P = 2.86 \times 10^{-7}$, Data Set C1). A *post hoc* Dunn test was also run to confirm the specific differences across sites, and all were significantly different from one another except Port Gamble and Skokomish (Data Set C1). This can be seen in Fig. 4 as the boxplots heavily overlap between these sites. Additional tests were performed on the log ratios to determine whether environmental variables also drove differences in these key taxa. Linear models were created to test the correlation between the log ratios of the above taxa and the mean values for temperature or dissolved oxygen over the 24 h prior to collection for each site and habitat. Neither of these linear models showed significant correlations of environmental conditions with the oyster-associated bacteria (linear regression correlation and *P* values: $R^2_{temperature} = -0.009721$, $p_{temperature} = 0.4709$; $R^2_{DO} = 0.02157$, $p_{DO} = 0.1557$).

In summary, Port Gamble and Skokomish experienced the highest overall mortality and highest fraction of *Vibrio*, Verrucomicrobiales, and *Synechococcus*.

## DISCUSSION

Olympia oysters in Puget Sound are a focal species for conservation and restoration science, due to the dramatic decline in population numbers from historical overfishing and failure of recovery efforts (1, 22). This field study found differences in Olympia oyster survival and microbiome between field sites, suggesting that some locations in Puget Sound may be more amenable to restoration than others. Temperature and dissolved oxygen were also significantly different across field sites. Upon further inspection, these variables only changed across sites and not between habitat types within those sites. There were also no differences within microbiome communities across the different habitats and no association between eelgrass habitat and oyster survival. The distance between eelgrass and unvegetated habitat at each site was minimal compared to geographic separation of the sites and leads to the conclusion that site characteristics were more impactful than microscale habitat changes.

Microbial communities showed significant variation across sample types: seawater, marine sediment, oyster shell biofilm, and oyster gut. The gut of the oyster hosted the lowest diversity of bacteria, which has been demonstrated previously in comparison to the surrounding water and sediment (23, 24). Beta diversity analysis suggests that the gut microbiome was significantly different from the microbiome found on the shell or in surrounding seawater. There are some shared ASVs between the gut and the surrounding environment, but these are primarily transient bacteria and the degree to which these bacteria are functional within the oyster gut is unclear. In another study, the biofilm of the shell of live and dead oysters was compared and found not to vary, suggesting that the shell microbiome is not controlled in the same way as the internal oyster tissue microbiota (23). Previous studies have demonstrated that the community of bacteria within the gut tends to be more controlled by the host itself than surrounding environmental variables (25). The ASVs unique to the oyster gut were, in fact, the most prevalent groups in the gut, creating a specialized microbial community. The oyster gut microbiome is hypothesized to break down polysaccharides and produce amino acids and vitamins, likely aiding in host digestion and nutrient absorption (26).

The most abundant bacteria within the oyster gut cannot be predicted by the environmental bacterial community or physical variables. In this study, *Mycoplasma* and an unidentified bacterial group made up a high percentage of the total community and were found in over 75% of oyster gut samples. *Mycoplasma* is a genus of the Mollicutes class and have been found in high proportions in various oyster species across a broad geographic range (12, 23, 24). One study demonstrated that *Mycoplasma* are likely relying on the oyster to provide certain compounds (27). The other highly abundant ASV in the oyster gut did not align to any known bacterial subgroups, which suggests some potential novelty in the microbiota of oysters. *Synechococcus* were also found in many of the oyster gut samples, and along with other cyanobacteria are frequently observed in the oyster gut (28, 29), but are likely sourced from the environment as they are also found frequently and in high proportions in seawater (28). While it is difficult to tease apart resident versus transient and

active versus inactive microbial populations from amplicon sequencing data, the groups identified here come to play an important role in further analysis.

Variation in gut microbiome composition by sites is largely driven by the balance of a few key taxa. Site-specific characteristics, such as temperature, salinity, and dissolved oxygen, may influence the abundances of these key taxa. In fact, many studies show significant dissimilarity in the internal oyster microbiome across growing locations (12, 14, 30). However, there is little evidence to suggest the gut bacteria originate solely from the environment (31). Some studies see far less variation in the microbiome across sites (15), but this could depend on how closely the sites are linked. The microbiome responds strongly to the food ingested by the oysters, and the type of food available is likely to change across habitats (32). In the case of this study, the variation can be summarized by the ratio of small groups of taxa across the sites. A higher ratio of *Vibrio*, Verrucomicrobiales, and *Synechococcus* in oyster gut microbiomes are responsible for the separation of Port Gamble and Skokomish from the other sites. Fidalgo Bay and Case Inlet, on the other hand, host more of the bacteria that are thought to be core to the oyster's gut tissue, particularly *Mycoplasma* spp. (12, 27, 33, 34). A previous study on Pacific oysters in the Hood Canal, Washington identifies Tenericutes (the phylum *Mycoplasma* belong to) and *Vibrio* in their samples, which matches the Hood Canal sites used in this study, Port Gamble and Skokomish (35). While *Vibrio* may be a common constituent of the oyster microbiome and are frequently non-pathogenic (36), they generally make up only a small percentage of the total community. In the case of Skokomish, *Vibrio* makes up a larger percentage than expected for a healthy oyster (Fig. 3). Port Gamble and Skokomish oysters also held a higher proportion of *Synechococcus* in their gut. *Synechococcus* could be a transient member of the community from filter-feeding, but previous studies also demonstrate its successful colonization of oyster tissue (29, 37). The overrepresentation of this taxa in Port Gamble and Skokomish oysters could indicate higher filtering activity than the oysters at Fidalgo Bay and Case Inlet or an increase in *Synechococcus* in the water column. The connection of Port Gamble and Skokomish by the Hood Canal supports the hypothesis that these sites experienced the same *Synechococcus* bloom.

The groups of bacteria which drive differences across sites also fluctuate similarly with respect to survival rate. Ratios of bacteria from Port Gamble and Skokomish were not statistically different from one another and these sites had the lowest survival rates (55% and 60%, respectively). Previous studies focused on stressed oysters suggest that proportions of *Vibrio* similar to those observed in our study are a sign of infection (38). Furthermore, the specific *Vibrio* sequence that is overrepresented in Skokomish aligns with species which have been reported as fish or shellfish pathogens, including *Vibrio toranzoniae*, *Vibrio crassostreae*, and *Vibrio kanaloae* (39–42). On the contrary, *Mycoplasma* is characterized as a core member of the oyster gut in this study and associated with higher survival. One study found that *Mycoplasma* actually increased in proportion in the gills of disturbed oysters (43), but as they are normally identified in the gut, this could be a sign of inappropriate translocation from the gut to more distal tissues, suggesting physiological disturbance. Therefore, the high prevalence of gut-associated *Mycoplasma* in our study is unlikely to be a sign of disturbance. In more recent years, research has explored the role of the microbiome in responding to viral or parasitic infections of oysters. Microbiome composition could be used to predict oyster mortality following exposure to OsHV-1, the ostreid herpesvirus (44). Susceptibility of oysters to infections could be dictated by their microbiome composition, stressing the importance of characterizing bacterial dynamics at oyster restoration sites. Oysters can also experience a loss of core bacteria following infection, such as the decreased abundance of Mollicutes in oysters infected with the parasite *Perkinsus marinus* (27). In this study, the reduced presence of *Mycoplasma* in the Skokomish samples could be explained by the increase in *Vibrio*, which may indicate some type of infection. The type of infection cannot be determined, as there are no records of *Vibrio* species causing disease in Olympia oysters. In anoxic conditions, the oyster microbiome may respond to host stress and shift toward an opportunist-dominated community, leading to mortality of the host, even if it had

the physiological capacity to withstand the anoxic conditions (11). Oysters at Skokomish were collected after a period of very low oxygen compared to the other sites, suggesting a stressful environment for the oysters and a likely cause for the dominance of potentially opportunistic *Vibrio* species in the gut microbiome at this site. While the microbiomes of dying oysters could not be captured in this study, the patterns between survival rate and bacterial differentials suggest a potential role of these bacteria in oyster mortality, which should be further tested.

Bacterial dynamics are important to consider when monitoring ecosystem health. A diverse set of microorganisms are better equipped to handle disturbance and outcompete invaders (25). Looking at the sites observed in this study, Fidalgo Bay varied greatly from Port Gamble and Skokomish, which are connected by the Hood Canal. The Fidalgo Bay oysters fared better than the Hood Canal oysters, which could predict higher likelihood of recruitment success and survival at Fidalgo Bay, compared with other sites. In fact, Fidalgo Bay restoration efforts have been very successful and native oyster populations grew from about 50,000 oysters in 2002 to almost 5 million in 2016 (45). Environmental conditions also varied in the time leading up to oyster collection, which could influence microbial communities in the environment and within the oyster. However, the environmental data failed to fully explain the variation in key bacterial taxa driving the differences across sites. There is no explanation yet as to why the bacterial communities varied so much or how to evaluate an optimal microbiome. Other variables that were not assessed in this study can also cause variation in the microbiome, such as estuary morphology (13), non-bacterial disease causing agents (46–48), and pollutants (49); it is possible that these other unknown variables may be linked to the oyster gut microbiota differences, and may be driving mortality rates. Transcriptional activity can also vary along environmental gradients and provide more insight about the behavior of bacteria within the oyster (50). While this type of data was not collected for this study, it will be an important factor to evaluate in the future.

As with any microbiome study, there are limitations in amplicon sequencing and deriving conclusions from a single time point of environmental data and tissues. Amplicon sequencing has biases in many steps of the process, from the initial subsampling of tissue to PCR primer bias. Bacterial proportions were not absolute, which prevents us from declaring that specific ASVs were increased or decreased from one sample to the next. However, future studies should aim for targeted quantification of the bacteria identified in this and other oyster microbiome studies or attempt to normalize ASV abundances with quantitative PCR of the total bacterial community (31). Moreover, microbiome data was only collected for one time point in the late summer. A time series of samples or an early sampling point for comparison may have revealed how the oyster microbiome initially responded to field conditions and how it changed over time. Temperature and dissolved oxygen variables were explored over the 24 h prior to collection, but the time of mortality for any lost oysters was unknown, meaning it was not possible to test association between these environmental conditions and mortality. Furthermore, it was not possible to statistically test correlation between survival rate and microbiome because the time of mortality for oysters at each site was unknown and microbiome of dying oysters was not captured. Additional constraints required all oysters to be held in one cage per site and habitat, which could lead to batch effects within the cages. Additionally, triplicate sediment and seawater samples were taken within close proximity of one another in order to investigate those communities closest to the oysters, but this likely led to higher similarity among the individual clusters and did not show a true range of alpha or beta diversity across the entire site. Considering such limitations, future field sampling efforts such as this should attempt to limit random and fixed effects as much as possible and collect widely dispersed samples to capture the full range of variation.

**Conclusions.** Oyster microbiomes have the potential to change because of their environment and/or host biology. This study demonstrated that while Olympia oyster gut microbiomes varied greatly by field site, the gut hosts a microbiome distinct from the surrounding environment. The microbial community was also associated with the survival rates, suggesting a connection between bacterial composition in the gut and

oyster performance. These outcomes have implications for restoration management of the native Olympia oyster in the Puget Sound, providing critical insight into the bacterial dynamics faced by oysters recruiting to these sites. Furthermore, this study takes one step toward developing microbiome analysis as a diagnostic tool, which could use oyster gut samples to determine whether a given population is under stress.

## MATERIALS AND METHODS

**Sampling.** Juvenile Olympia oysters (~1 year old) were collected from the hatchery at the Kenneth K. Chew.

Center for Shellfish Research in Manchester (WA, USA) and distributed to four field sites throughout Puget Sound in June of 2018 and retrieved 2 months later in August 2018 (Fig. 1). All oysters used were from a common genetic background (a subpopulation of Fidalgo Bay oysters) and were raised in the same hatchery conditions. At each of the four field sites, one PVC mesh oyster cage was deployed in the center of a patch of eelgrass (*Zostera marina*) habitat and another cage was deployed in the center of a patch of unvegetated habitat. The 1-cm mesh-size cages were intended to exclude predators while allowing circulation. Each cage was anchored to a PVC post and contained 10 oysters upon deployment. A "patch" of eelgrass habitat was defined as an area at least 6 m in diameter with at least 60 shoots per square meter, and a "patch" of unvegetated habitat was defined as an area at least 6 m in diameter with no eelgrass present. The centroid of all patches was located at a tidal elevation between −0.3 m and −1 m MLLW. Cages were cleaned of biofouling organisms and debris every 2 weeks during the deployment.

Upon retrieval, three water samples and three sediment samples were taken from the area around each oyster cage (*n* = 6 water and 6 sediment samples per site). At Case Inlet, only three water samples were taken (*n* = 2 inside eelgrass beds and *n* = 1 outside eelgrass) due to a shortage of bottles in the field. Water samples were collected within 3 m of each oyster cage on an ebbing tide, when the water column was approximately 1 m deep. Samples were collected in acid-washed Nalgene bottles with mesh filters over the opening. The bottle was dipped below the surface of the water while wearing gloves and kept underwater until nearly full. Sediment samples were collected in 15 mL Falcon tubes by opening the tubes at the top of the sediment, sweeping the tube opening across the top 1 in. of sediment and then pouring out excess water before capping. Oyster cages were then retrieved and transported to the laboratory within 1.5 to 2 h in cool, dark and dry conditions.

In the laboratory at the University of Washington, oyster shells were lightly scrubbed with sterile toothbrushes to remove mud and left to dry for a few minutes. Biofilm samples were collected from three oysters in each cage by swabbing back and forth across the entirety of the shell surface on one side. Swab tips were removed, placed in individual 1.5 mL vials, immediately frozen in a dry ice bath, and then stored at −80℃. Shell length was recorded for all oysters after swabbing to prevent cross contamination. Living oysters were then shucked using a sterile scalpel. Complete stomach and digestive tissue were removed using a newly sterilized scalpel blade, flash frozen, and then stored at −80℃. For each oyster cage, survival was recorded as the proportion of living oysters remaining out of 10.

Sediment samples were stored at −80℃ upon arrival at the laboratory, and water samples were filtered over 0.2 $\mu$m-pore size cellulose filters using vacuum filtration. The filters were folded and dropped into Powerbead tubes from the Qiagen DNeasy Powersoil Kit and stored at −80℃.

**Environmental data collection.** PME miniDOT sensors (for temperature and dissolved oxygen data) and Odyssey conductivity loggers (for salinity data) were deployed alongside oyster cages in eelgrass habitat and in unvegetated habitat at each site. Instruments logged at 10-min intervals from early June 2018 to late August 2018. Measurements collected when the predicted tidal elevation was lower than 0 m MLLW were excluded to eliminate data collected during immersion. Dissolved oxygen data were adjusted based on salinity and reported in mg*L$^{-1}$.

To assess relative differences between habitats and between field sites, temperature and dissolved oxygen data from the 24 h immediately prior to collection were analyzed. A permutational two-way ANOVA for repeated measurements was run to account for repeated measures from the same sensors at the same sites over time (51). This data did not follow a normal distribution, and therefore the permutational ANOVA approach was used. The interaction between site and habitat was also explored when assessing differences in the environmental data.

**DNA extraction, amplification, and sequencing.** Following the Earth Microbiome Project protocols, DNA was extracted from all sample types using the single tube Qiagen DNeasy Powersoil Kit. Single tube extractions, although more time-consuming, reduce the amount of well-to-well contamination (52). Extracted DNA was shipped over dry ice to Scripps Institution of Oceanography and stored at −20℃. DNA was amplified following the 16S rRNA gene Illumina amplicon protocol provided by the Earth Microbiome Project (53). Primers 515F and 806R were used to target the V4 region of the 16S rRNA gene and sequenced on the Illumina MiSeq platform to produce 250 bp forward and reverse reads.

**Sequence analysis.** Resulting sequence data were uploaded to Qiita (54) (Qiita ID 12079) and demultiplexed, trimmed to 150 bp and erroneous sequences were removed using the Deblur workflow positive filter (55). The deblur final table was exported to Qiime2 (56) and used for all subsequent analyses. Alpha diversity across sample types was assessed by Shannon diversity index (57), which measures richness and evenness within given sample types (Fig. 2). Significance of alpha diversity across groups was conducted with a Kruskal-Wallis test. Beta diversity was analyzed via Bray Curtis (58), weighted and unweighted UniFrac (59, 60), and Qiime2's DEICODE RPCA (61) method with a sampling depth of 1,920. A sampling depth of 1,920 was chosen based on rarefaction curves, which are displayed in the

supplementary data (Data Set F). The number of observed ASVs started to plateau around 2,000 sequences, but in order to retain one feature which had 1,922 sequences, the sampling depth used was 1,920. Phylogenetic tree derivation for UniFrac was performed using an insertion tree with the fragment insertion sepp function in Qiime2 (62). PERMANOVA tests for all beta diversity metrics were run in Qiime2 (63). RPCA was chosen for presentation because this method does not use pseudocounts and is therefore termed a more robust version of the Aitchison's distance metric (Fig. 2). Taxonomy was assigned in Qiime2 against the Silva database v.138 (64, 65). The biom table and taxonomy was downloaded from Qiime2 and reconstructed in R using the program Qiime2R. The taxonomy bar plots and heat maps were generated in R (Fig. 3), alongside the alpha diversity boxplot in Fig. 2. All samples that were retained through the Deblur workflow are presented in the taxonomy plots in Fig. 3. The heatmap encompasses sediment, water, biofilm, and oyster gut samples, while the bar plot was generated using only oyster gut samples. RPCA analysis was conducted once again, but after filtering out all sediment, seawater, and shell biofilm samples to include oyster gut samples only. The purpose of this was to further investigate the differences within oysters across field sites (Fig. 4). For oyster gut samples, DEICODE RPCA beta diversity analysis was performed at a sampling depth of 1,000 because the alpha diversity started to plateau at a lower sequencing depth compared to other sample types (Data Set F). This depth allowed one additional gut sample to be retained in the analysis. Songbird differential abundance analysis was then performed to rank the differentials of every ASV across field sites (66).

**Data availability.** Sequence data generated in this project are deposited in the EBI-ENA database and NCBI BioProject database under (accession no. PRJEB49367) and made available through Qiita (Study ID: 12079). Processed data files and scripts for Qiime2 and R are available in the GitHub Repository (https://github.com/ekunselman/OlympiaOysterMicrobes).

## SUPPLEMENTAL MATERIAL

Supplemental material is available online only.
**SUPPLEMENTAL FILE 1**, PDF file, 0.3 MB.

## ACKNOWLEDGMENTS

Thank you to Laura H. Spencer for providing the oyster spat used in this study. Thank you to the Ryan Kelly lab for allowing me to use their lab space for dissection of oysters and extractions. Thank you to the Washington Department of Natural Resources for supplying extraction kits.

This project was supported by the US National Science Foundation grant OCE-1837116 to E.E.A. and funding from the Aquatic Assessment and Monitoring Team at the Washington State Department of Natural Resources. J.J.M. was supported by NSF postdoctoral research fellowship in biology award number 2011004.

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
