## [Reviewer comments · Microbiology Spectrum]

Microbiology Spectrum

Variation in survival and gut microbiome composition of hatchery grown native oysters at various locations within the Puget Sound

Emily Kunselman, Jeremiah Minich, Micah Horwith, Jack Gilbert, and Eric Allen

Corresponding Author(s): Emily Kunselman, Scripps Institution of Oceanography

Review Timeline:

Submission Date:	October 19, 2021
Editorial Decision:	December 3, 2021
Revision Received:	January 28, 2022
Editorial Decision:	February 17, 2022
Revision Received:	April 16, 2022
Accepted:	April 17, 2022

Editor: Konstantinos Kormas

Reviewer(s): Disclosure of reviewer identity is with reference to reviewer comments included in decision letter(s). The following individuals involved in review of your submission have agreed to reveal their identity: Timothy J Green (Reviewer #3)

Transaction Report:

DOI: <https://doi.org/10.1128/Spectrum.01982-21>

December 3, 2021

Dr. Emily Kunselman
Scripps Institution of Oceanography
8750 Biological Grade
La Jolla, CA

Re: Spectrum01982-21 (Survival of hatchery grown native oysters is associated with specific gut-associated bacteria at various locations within the Puget Sound)

Dear Dr. Emily Kunselman:

Link Not Available

Sincerely,

Konstantinos Kormas

Journals Department
Reviewer comments:

Reviewer #1 (Public repository details (Required)):

I would suggest submission of sequencing reads to NCBI in addition to proposed repository. Data sharing plan is adequate.

Reviewer #1 (Comments for the Author):

Stats seem appropriate for the microbiome analysis, but some more detail about the results from the statistical tests should be included in the manuscript.

Reviewer #2 (Public repository details (Required)):

16S rRNA gene sequences should be deposited in a public repository. The authors have stated "Sequence data generated in this project will be deposited in the EBI-ENA database and made 430 available through Qiita (Study ID: 12079)".

Reviewer #2 (Comments for the Author):

The authors have already highlighted limitations to this study eg lack of time points, batch effects, lack of quantitative data. One approach to deal with the lack of absolute abundance data which could be further discussed, would be to normalise the data against total 16S bacterial qPCR values (eg King WL, et al. (2021). *Front Microbiol* 12:723649). While this is not absolute quantification, it would allow comparisons of specific taxa abundance between samples.

In addition, future work could be directed at specifically detecting bacterial groups identified as being of interest such as *Mycoplasma*, *Vibrio* and *Synechococcus*, for example, by qPCR, in addition to whole bacterial community analysis.

Reviewer #3 (Public repository details (Required)):

Manuscript indicates files are deposited in a public repository

Reviewer #3 (Comments for the Author):

This manuscript investigated the bacteria community of the Olympia oyster in four locations of Puget Sound. The major finding is the bacteria community of the digestive gland of oysters differs from the surrounding seawater, sediment and oyster shell. These results are interesting to the field of oyster health and restoration, but not novel. The manuscript is well written, conclusions are strongly supported by the results with appropriate methods and statistical analysis. Limitations are identified and clearly discussed.

The manuscript could be improved if the authors could provide other metrics for oyster health, such as growth rate, condition or presence of biofouling. I was a little confused with some of the environmental data and its interpretation due to the faulty salinity measurements from malfunctioning instruments. It would also be interesting if the *Vibrio* associated with higher mortality of oysters is a single ASV (potentially indicating an infection). I also appreciate that the authors may have decided not to speculate because of the close 16S rRNA gene sequences for the genus.

I am assuming the reported oxygen in mg.l was not corrected for salinity (line 132). The increased proportion of *Mycoplasma* in oysters from Case Inlet and Fidalgo Bay may also be because these oysters were not feeding at the time of sampling, which might be supported by reduced proportion of *Synechococcus* (transient bacteria from feeding??).

Line 24 - biparental family? Or was it a mix of families from a single spawn.

Line 85 - please provide additional information about genetics of oysters. Large amount of recent research has shown that genetics influences the bacterial community.

Line 107 - the amount of time that oysters were transported between sites and would this influence the bacteria community (i.e. 1 hour for site A and 4 hours for site D).

Line 161 - sequencing depth of 1,920. Is this not low for microbiome analysis, which is typically >10,000 reads per sample.

Line 178 - please provide statistical justification for increased survival. Maybe not possible with 1 replicate cage.

Line 240 - abundance or proportion of ASVs (amplicon reads).

Line 334 - same as line 240. Could the abundance of *Vibrio* increased in a sick oyster, and so the proportion (%) of core bacteria in the community reduced. Would performing a qPCR assay for total 16S rRNA and *Vibrio* 16S rRNA provide data on this point.

Line 363 - minor grammar "(29)".

Staff Comments:

Preparing Revision Guidelines

Please return the manuscript within 60 days; if you cannot complete the modification within this time period, please contact me. If you do not wish to modify the manuscript and prefer to submit it to another journal, please notify me of your decision immediately so that the manuscript may be formally withdrawn from consideration by Microbiology Spectrum.

Spectrum 01982-21 review

This study characterizes the microbiome of Olympia oysters after a field study at 4 location in Puget Sound. The study was designed to evaluate the effect of site on survival and microbiome composition in oysters, with the goal of informing oyster restoration. As such, it has a strong rationale and justification. The study provides a useful baseline of knowledge about differences in microbial communities in Olympia oysters between sites. The microbiome analysis methods and the sample size used (when pooling samples from 2 habitats within site) are valid for determination of differences in microbiome composition between sites.

The manuscript, however, as currently written, has somewhat overstated the conclusions and implications of the work by strongly focusing on role of microbiome on survival (as illustrated in the title). There are some issues with the experimental design that preclude any strong conclusions about a relationship between survival at a site and microbiome composition. These include: a) there was no true cage replication at each site, so the effect of site on survival could not be statistically determined; and b) No diagnosis was performed on the oysters, and the timing of mortality for each cage is not known. As such, the discussion has several areas in which the conclusions are not necessarily supported by the data. For example, there is no evidence from the presented data indicating dysbiosis at any of the oysters or sites. Regarding the role of vibrios on survival, not all vibrio spp. present in oyster gut samples are pathogenic - indeed, most vibrio spp. are non-pathogenic. Do vibrios cause disease outbreaks in Olympia oysters, or are this just due to secondary growth of bacteria due to other disease issues?

Some more detailed comments:

Line 30 - This conclusion about habitat type may be overstated - was there enough power in the experiment to detect differences between habitat site, just with 10 oysters in one single cage?

Line 62 - Not sure what it is meant by "as microbes are either exposed to the same environmental conditions or are...." in regards to the impact of stress on host microbiomes.

Line 70 - In addition to affecting the host through production of antibiotics, the microbiome can potentially affect the host health through a variety of mechanisms, from improving digestion and providing nutrients, detoxification, modulation of inflammation, etc. I suggest adding here some general literature from other organisms and oysters (e.g. probiotics).

Line 88 - only one cage per site - no replication at each site.

Line 162 - Sampling depth of 1920 for RPCA - why? Could you provide a rationale?

Line 174 - The sample depth of 1000 reads was used to perform the analysis with the gut samples, which is a pretty low number of reads. Authors should provide evidence from a rarefaction analysis or other analysis that this sample depth provides a true representation of the community at each site. Authors should also show as supplementary data the read depth for each oyster and site and sample type, so the reviewers can assess any potential biases due to major differences in read depth between these factors.

Results

Line 180 (and Figure 1)- The text mentions differences in survival between habitats, is this the average for all sites? If so, provide average and standard deviation, and if that difference in survival is consistent at all sites. In figure 1, the data for survival should be shown for each cage, by site and habitat, or, at the very least, the mean survival plus/minus standard deviation for the two cages at each site.

Line 200 - Figure 2A shows very similar values of Shannon index (overlapping) between biofilm and marine sediment, are these truly significant in pairwise comparisons? Please show overall stats as supplementary data.

Line 203 - Indicate in the text that this data is shown in Fig 2A (not just figure 2). Same for the following lines (indicate which of the panels within Fig 2 show the data corresponding to each sentence).

Lines 226 - 270: Much of this information belongs in the methods section.

Authors should also show data on alpha and beta diversity by site for the gut

samples (needed to make any arguments about dysbiosis).

Figure 1 - What is the process of data curation for DO and Temp? There seems to be some dips in DO in the CL eelgrass that are not seen at any of the other sites. Are these real?

Figure 2. Clarify in the figure legend that this is data is for only 3 oysters per cage. For 2A indicate significance for pairwise comparisons in the legend or in the figure. For 2B - I recommend that the authors use a combination of symbols (open and closed) to indicate the site and habitat source for each oyster., and better define in the legend what the figure is showing, which groupings are significantly different, and provide a table with the loadings for each grouping.

Conclusion

Lines 279 - 280 - I would not base the choice of restoration sites based on this limited set of data. The experimental design does not allow to establish a relationship between particular taxa and survival.

Lines 363 - 367 - These statements are inaccurate, are not a reflection of what is known about oyster diseases in general or disease in *Olympia* oysters in particular. For example, what do the authors mean that "pathogens can come from within the oyster"?

Throughout - Use a consistent format for int text citations.

Response to Reviewers

Reviewer #1

Reviewer #1 (Public repository details (Required)):

I would suggest submission of sequencing reads to NCBI in addition to proposed repository. Data sharing plan is adequate.

Reads submitted to ENA are deposited into the SRA, and so they are available on NCBI - (Project Accession: PRJEB49367)

Reviewer #1 (Comments for the Author):

Stats seem appropriate for the microbiome analysis, but some more detail about the results from the statistical tests should be included in the manuscript.

Tables (Supplementary tables A, B, C, D) for relevant statistical tests have been added to the supplementary data and all formulas are available in the script files of the public GitHub repository "Olympia Oyster Microbes". Most tests used to calculate significance are non-parametric tests due to the nature of the data (microbiome data uses sparse matrices which are not normally distributed).

This study characterizes the microbiome of Olympia oysters after a field study at 4 location in Puget Sound. The study was designed to evaluate the effect of site on survival and microbiome composition in oysters, with the goal of informing oyster restoration. As such, it has a strong rationale and justification. The study provides a useful baseline of knowledge about differences in microbial communities in Olympia oysters between sites. The microbiome analysis methods and the sample size used (when pooling samples from 2 habitats within site) are valid for determination of differences in microbiome composition between sites.

The manuscript, however, as currently written, has somewhat overstated the conclusions and implications of the work by strongly focusing on role of microbiome on survival (as illustrated in the title). There are some issues with the experimental design that preclude any strong conclusions about a relationship between survival at a site and microbiome composition. These include: a) there was no true cage replication at each site, so the effect of site on survival could not be statistically determined; and b) No diagnosis was performed on the oysters, and the timing of mortality for each cage is not known.

These concerns are valid, and so as not to mislead the reader, the title has been changed. We have made efforts to make it clear in the main text that these are only observed trends and we do not have the power to statistically connect the microbiome to survival (lines 386-388). These limitations have been further exemplified in the text (Lines 417-429).

As such, the discussion has several areas in which the conclusions are not necessarily supported by the data. For example, there is no evidence from the presented data indicating dysbiosis at any of the oysters or sites.

While the shift in Vibrio composition and loss of Mycoplasma at some sites are implicated in the literature as signs of dysbiosis (DOI:10.3389/fimmu.2021.630343), the word itself can have different meanings to different people, and therefore we have removed it from the text and opted for alternative language.

Regarding the role of vibrios on survival, not all vibrio spp. present in oyster gut samples are pathogenic - indeed, most vibrio spp. are non-pathogenic. Do vibrios cause disease outbreaks in Olympia oysters, or are this just due to secondary growth of bacteria due to other disease issues?

We have mentioned in the text that Vibrios are “a common constituent of the oyster microbiome and are generally non-pathogenic” so as not to mislead the reader (lines 346-347). While we cannot determine with confidence that the species present in Skokomish are pathogenic, the overabundance of Vibrio could suggest opportunistic or pathogenic behavior. However, as you point out, there are no studies that identify Vibrio infections of Olympia oysters (specifically Vibrio that are harmful to the oyster rather than human Vibrio pathogens accumulating in the oysters). This caveat has been added to the discussion (lines 379-380).

Some more detailed comments:

Line 30 - This conclusion about habitat type may be overstated - was there enough power in the experiment to detect differences between habitat site, just with 10 oysters in one single cage?

For these tests, there were 31 oysters from eelgrass and 27 oysters from unvegetated habitat across all the sites. There were also 12 sediment, 12 water and 12 shell biofilm samples for each habitat type. This brings the total to around 65 samples per habitat type. There was no interaction between site and habitat or sample type and habitat. For both Alpha and Beta diversity statistical tests, F was very low, and the p value was very high, so we do not expect that higher sample sizes would yield a different result.

Line 62 - Not sure what it is meant by “as microbes are either exposed to the same environmental conditions or are....” in regards to the impact of stress on host microbiomes.

Apologies, we can see that this was poorly worded, and have adjusted the sentence to make it clearer. It now says, “Environmental stress can alter diversity and composition of oyster microbiomes, either as a result of bacterial response to the changing environment, or to the host’s changing gene expression” (lines 57-59). For example, the host immune response could be changing by increasing or decreasing antimicrobial activity.

Line 70 - In addition to affecting the host through production of antibiotics, the microbiome can potentially affect the host health through a variety of mechanisms, from improving digestion and providing nutrients,

detrification, modulation of inflammation, etc. I suggest adding here some general literature from other organisms and oysters (e.g. probiotics).

This is a great addition – we have added a list of bacterial contributions to their host and corresponding literature, all of which derived from invertebrate models.

Line 88 - only one cage per site - no replication at each site.

This was identified as a study constraint. After finding no differences in the microbiome between eelgrass and bare habitat, we technically have 2 cages per site.

Line 162 - Sampling depth of 1920 for RPCA - why? Could you provide a rationale?

An initial rarefaction depth of 1920 was selected because the rarefaction curve shows that Shannon alpha diversity plateaus for all sample types after ~2000 sequences. However, gut samples have much lower sequence counts than other samples, and in order to retain most of the gut samples, we must rarefy below 2000 sequences. The specific count of 1920 was chosen based on the feature counts per sample after filtering the OTU table to remove chloroplast, mitochondrial, and other reads only present 3 or fewer times. The feature count which was closest to 2000 was 1922 for PG Gut eelgrass 11. Therefore, this sample was retained in the analysis by rarefying at 1920 for the alpha diversity analysis and RPCA beta diversity analysis. All feature counts lower than this were below 1500 sequences.

Line 174 - The sample depth of 1000 reads was used to perform the analysis with the gut samples, which is a pretty low number of reads. Authors should provide evidence from a rarefaction analysis or other analysis that this sample depth provides a true representation of the community at each site. Authors should also show as supplementary data the read depth for each oyster and site and sample type, so the reviewers can assess any potential biases due to major differences in read depth between these factors.

The rarefaction depth was decreased when only looking at gut samples. The reason for this was because mean oyster sample alpha diversity plateaued around 1000 reads, in comparison to the other samples which held higher richness of OTUs. One additional gut sample could be retained for the analysis by rarefying at 1000. Rarefaction curves have been included in the supplementary data for observed OTUs and Shannon's index across sample types and by individual oysters from each site (Supplemental Figure F).

Results

Line 180 (and Figure 1)- The text mentions differences in survival between habitats, is this the average for all sites? If so, provide average and standard deviation, and if that difference in survival is consistent at all sites. In figure 1, the data for survival should be shown for each cage, by site and habitat, or, at the very least, the mean survival plus/minus standard deviation for the two cages at each site.

Yes, this is the average for all sites. We have specified this and added the standard deviations for each habitat in the text and provided 'per cage' survival and sample sizes in the figure by dividing them between habitats at each site. The difference in survival between habitat types is not significant and varies by site, with some sites having higher survival in eelgrass (Case Inlet & Skokomish) and some sites having higher survival in the bare habitat (Port Gamble and Fidalgo Bay). We have included the p-value from a two-way t-test to make this point clearer.

Line 200 - Figure 2A shows very similar values of Shannon index (overlapping) between biofilm and marine sediment, are these truly significant in pairwise comparisons? Please show overall stats as supplementary data.

For Shannon alpha diversity group significance, all groups are significantly different from one another. The Kruskal -Wallis pairwise comparisons have been included in the supplementary data and this shows that, despite proximity in the figure, even sediment and biofilm groups vary significantly in their alpha diversity (Supplementary Figure E). We have also added significance bars in the figure 2A itself to indicate all groups are different from one another.

Line 203 - Indicate in the text that this data is shown in Fig 2A (not just figure 2). Same for the following lines (indicate which of the panels within Fig 2 show the data corresponding to each sentence.

Thank you for the recommendation. We have updated all figure references to say both figure number and panel.

Lines 226 - 270: Much of this information belongs in the methods section. Authors should also show data on alpha and beta diversity by site for the gut samples (needed to make any arguments about dysbiosis).

We have transferred some of the methodology information into the methods section, but the remaining discussion of Qurro, Songbird, DEICODE, etc. are important for understanding the analysis and results in entirety. Therefore, we have chosen to keep them with the results. Alpha diversity within the oyster guts does not significantly vary across sites. We have included a statement on alpha diversity in lines 231-232. For this reason, most of the analysis focuses on compositional differences between the sites. Beta diversity by site within the gut samples is illustrated in Figure 4 by the RPCA PCoA plot and statistics for significant differences are reported in the text.

Figure 1 - What is the process of data curation for DO and Temp? There seems to be some dips in DO in the CL eelgrass that are not seen at any of the other sites. Are these real?

Data collection and curation was performed as follows, which is detailed in the methods (lines 128-133): "Measurements collected when the predicted tidal elevation was lower than 0 m MLLW were excluded to eliminate data collected during emersion. Dissolved oxygen data were adjusted based on salinity and reported in $\text{mg}\cdot\text{L}^{-1}$. To assess relative differences

between habitats and between field sites, temperature, and dissolved oxygen data from the 24 hours immediately prior to collection was analyzed.”

Apart from the exclusion of data from periods of predicted emersion, all environmental data are visualized in Fig 1B and included in statistical comparisons to avoid arbitrary identification of ‘good’ and ‘bad’ data. The dips in DO at CI Eelgrass could reflect natural phenomena. It is common for piles of wrack to accumulate in eelgrass meadows in late summer, as shoots shed senescent leaf material and the meadow captures other detritus due to attenuation of flow within and above the canopy. Microbial respiration in these wrack piles can cause local depletion of dissolved oxygen as biomass breaks down. It is also possible that the sensor at CI Eelgrass malfunctioned during this period, due to fouling or sedimentation on the sensing membrane or other issues, but the subsequent convergence of DO values at CI Eelgrass and CI Bare appears to suggest that both sensors were functional, and that the period of low DO at CI Eelgrass reflects environmental conditions. There was no evidence of biofouling or damage on the dissolved oxygen sensor at CI Eelgrass when it was retrieved.

Figure 2. Clarify in the figure legend that this is data is for only 3 oysters per cage. For 2A indicate significance for pairwise comparisons in the legend or in the figure. For 2B - I recommend that the authors use a combination of symbols (open and closed) to indicate the site and habitat source for each oyster., and better define in the legend what the figure is showing, which groupings are significantly different, and provide a table with the loadings for each grouping.

For 2A, we have updated the legend to specify the “n” per cage. We added bars to demonstrate pairwise significance across all groups for alpha diversity. For the PCA plot in 2B, we have decided that using different shapes and symbols for both habitat type and geographic site is too overwhelming for the figure. Since habitat type has no significant effect on the microbiome, we are only adding a new legend and shapes to differentiate geographic site. We have updated the figure description with more detail. RPCA is primarily concerned with feature (ASV) loadings rather than sample loadings, so the loadings for each sample type grouping are not readily available. In addition, Qurro can be used to visualize feature loadings, but you must view these loadings as a ratio between ASVs. We have corrected a portion of text to explain that each sample grouping is significantly different from one another based on pairwise comparisons of each sample type.

Conclusion

Lines 279 - 280 - I would not base the choice of restoration sites based on this limited set of data. The experimental design does not allow to establish a relationship between particular taxa and survival.

The study does not claim statistical power to confirm the link between particular taxa and survival. However, the differences in survival and microbiomes, even on their own, are partial indicators of suitability for restoration.

Lines 363 - 367 - These statements are inaccurate, are not a reflection of what is known about oyster diseases in general or disease in Olympia oysters in particular. For example, what do the authors mean that “pathogens can come from within the oyster”?

We apologize that the wording has confused the reader. Rather than describing oyster disease, the goal is to explain that more information about the role of the microbiome in disease is being published. We hope to rationalize some of the patterns that were seen in the data by comparing our outcomes to other studies that characterize the microbiome in response to disease or severe stress. We have chosen to remove the text that pathogens can come from within the oyster, because it is better described as a shift towards opportunistic species (which may already be present in the oyster but are controlled by the host immune response and antibiotic activity of other bacteria).

Throughout - Use a consistent format for int text citations.

We have ensured that citations follow a consistent pattern.

Reviewer #2

The manuscript describes differences in the microbial community from oyster gut, sediment and seawater in relation to habitat (eelgrass vs no eelgrass), site and physicochemical parameters. The study found significant differences in oyster survival and microbiome across sites, but not between habitats. Further analysis of the oyster gut microbial community was performed to identify core bacterial taxa and link the presence of specific bacterial taxa (eg *Mycoplasma*, *Vibrio*, *Synechococcus*) with oyster survival rates. The results will assist in identifying suitable sites for Olympia oyster restoration and the microbiome approach may be used to assess oyster health. The manuscript is well written and makes a significant contribution to the scientific community.

Reviewer #2 (Public repository details (Required)):

16S rRNA gene sequences should be deposited in a public repository. The authors have stated "Sequence data generated in this project will be deposited in the EBI-ENA database and made 430 available through Qiita (Study ID: 12079)".

Reviewer #2 (Comments for the Author):

The authors have already highlighted limitations to this study eg lack of time points, batch effects, lack of quantitative data. One approach to deal with the lack of absolute abundance data which could be further discussed, would be to normalise the data against total 16S bacterial qPCR values (eg King WL, et al. (2021). *Front Microbiol* 12:723649). While this is not absolute quantification, it would allow comparisons of specific taxa abundance between samples. In addition, future work could be directed at specifically detecting bacterial groups identified as being of interest such as *Mycoplasma*, *Vibrio* and *Synechococcus*, for example, by qPCR, in addition to whole bacterial community analysis.

Unfortunately, we are unable to run all the suggested qPCR for these samples. However, we agree that future work should target the quantification of these taxa of interest. We have added this suggestion in the discussion with the citation for reference (lines 414-417).

Reviewer #3

Reviewer #3 (Public repository details (Required)):

Manuscript indicates files are deposited in a public repository

Reviewer #3 (Comments for the Author):

This manuscript investigated the bacteria community of the Olympia oyster in four locations of Puget Sound. The major finding is the bacteria community of the digestive gland of oysters differs from the surrounding seawater, sediment and oyster shell. These results are interesting to the field of oyster health and restoration, but not novel. The manuscript is well written, conclusions are strongly supported by the results with appropriate methods and statistical analysis. Limitations are identified and clearly discussed.

The manuscript could be improved if the authors could provide other metrics for oyster health, such as growth rate, condition or presence of biofouling.

Unfortunately, the length of the oysters could not be measured at the beginning of the field exposure, so we do not have growth data. Additionally, we do not have any weight measurements (of tissue or whole shell). We do have the lengths of the oysters at the end of the experiment. The lengths do vary significantly across the sites (single factor ANOVA, $p = 0.006$ for shell width, $p = 0.038$ for shell length), but we do not think this is worth reporting because we do not have the change in length, and therefore cannot make any assumptions about differences in growth across sites. We do not have qualitative metadata for biofouling, but all the "Cages were cleaned of biofouling organisms and debris every two weeks during the deployment" (lines 95-96) as a protective measure for the oysters.

I was a little confused with some of the environmental data and its interpretation due to the faulty salinity measurements from malfunctioning instruments.

At Fidalgo Bay Eelgrass, Case Inlet Eelgrass and Case Inlet Bare, the Odyssey Conductivity Sensors failed due to water intrusion and/or severe fouling between the electrodes. Because of this data gap, salinity time series data were not analyzed or considered. Salinity was used for one purpose only in our manuscript: to adjust dissolved oxygen data. We followed manufacturer protocols and used the program 'miniDOT Concatenate Rev 4.04' to adjust dissolved oxygen data after instruments were retrieved. This program uses a single salinity value (in ppt) to adjust the entire time series of dissolved oxygen data. For sites where Odyssey Conductivity Sensors remained functional, we used the mean salinity value from the deployment (32.6 for PGB, 33.0 for PGE, 31.9 for FBB, 31.7 for SKB, and 30.8 for SKE). For sites where the Odyssey Conductivity Sensors failed, we used the

mean salinity value from the adjoining site. For example, the in situ value of 31.9 for FBB was used for FBE. At CI, where Odyssey Conductivity Sensors failed at both the Eelgrass site and the Bare site, we obtained salinity data from the Washington State Department of Ecology Long-Term Marine Water Monitoring Program, which recorded a surface salinity of 29.5 in Case Inlet (station CSE001) in August 2018.

It would also be interesting if the *Vibrio* associated with higher mortality of oysters is a single ASV (potentially indicating an infection). I also appreciate that the authors may have decided not to speculate because of the close 16S rRNA gene sequences for the genus.

The Vibrio identified in the RPCA biplot as driving differences between sites is a single ASV. However, the songbird differentials include all ASVs from the gut samples that were identified as Vibrio at the genus level. This comprises 2 ASVs. The primary Vibrio ASV driving the separation for Skokomish does not have a singular species hit when BLAST against NCBI nucleotide database, but the top hits do comprise a list of known or suspected oyster pathogens (1- Vibrio toranzoniae, 2- Vibrio crassostreae, and 3- Vibrio kanaloae). However, we cannot statistically associate these with mortality due to a lack of power.

I am assuming the reported oxygen in mg.l was not corrected for salinity (line 132).

Lines 129-130 state that "Dissolved oxygen data were adjusted based on salinity and reported in mg*L⁻¹" All dissolved oxygen data were corrected for salinity using methods and values detailed above.

The increased proportion of *Mycoplasma* in oysters from Case Inlet and Fidalgo Bay may also be because these oysters were not feeding at the time of sampling, which might be supported by reduced proportion of *Synechococcus* (transient bacteria from feeding??).

This is a great point. We have adjusted the discussion to include this point before moving on to the dynamics between Mycoplasma and Vibrio.

Line 24 - biparental family? Or was it a mix of families from a single spawn.

A group of Olympia oysters originally from a North Fidalgo Bay subpopulation were maintained in a research facility in Manchester, Washington and conditioned to spawn. The spawn were raised in a common tank and oysters from this spawn were outplanted in all the field sites for this study. In the manuscript, we have included a line to indicate this: "All oysters used were from a common genetic background (a subpopulation of Fidalgo Bay oysters) and were raised in the same hatchery conditions."

Line 85 - please provide additional information about genetics of oysters. Large amount of recent research has shown that genetics influences the bacterial community.

A line was included in the methods to explain that all oysters used in this study were from the same genetic background and raised under the same conditions.

Line 107 - the amount of time that oysters were transported between sites and would this influence the bacterial community (i.e. 1 hour for site A and 4 hours for site D).

All locations were a similar distance from the lab (1.5-2 hours), so it is unlikely that this would have driven any microbial differences between the sites. This is a good point and if there were greater differences in travel time, the oysters likely would have been dissected and frozen in the field.

Line 161 - sequencing depth of 1,920. Is this not low for microbiome analysis, which is typically >10,000 reads per sample.

Sequencing depth may be considered low in comparison to other microbiome analyses. In this study, we especially had to account for the lower diversity exhibited by oyster guts. Some (not all) prior literature shows similar rarefaction for oysters/ oyster guts (Dubé, Ky, and Planes 2019; Trabal Fernandez et al 2014). We have added the rarefaction curves to the supplementary data to demonstrate that the diversity plateaus early, justifying the ability to rarefy at 1,920 for all sample types, and at 1,000 for gut samples only.

Line 178 - please provide statistical justification for increased survival. Maybe not possible with 1 replicate cage.

We ran a proportion test in R on the number of surviving oysters for each site (for both eelgrass and bare habitat included) out of the total oysters at that site, and it reported that survival was significantly different with a p value of 0.0258. However, after running a pairwise proportion test, only the difference between Port Gamble and Case Inlet is significant. We are reporting this in the text for clarification. However, as explained in the limitations of the study, this survival cannot be directly linked to the microbiome since we do not know the microbiome at the time of death. The survival data in tandem with specific bacteria in the gut helps generate hypotheses about the types of bacteria to monitor in future studies.

Line 240 - abundance or proportion of ASVs (amplicon reads).

We have tried to explain the “zeros” in a different way. If an ASV that was not detected in a sample, then its value is zero, and so the log ratio cannot be computed.

Line 334 - same as line 240.

We have decided to stick with “ratio”, as this is the most accurate way to explain the comparison between the taxa.

Could the abundance of Vibrio increased in a sick oyster, and so the proportion (%) of core bacteria in the community reduced. Would performing a qPCR assay for total 16S rRNA and Vibrio 16S rRNA provide data on this point.

We agree this is the hypothesis. We have highlighted the point that core bacteria (Mycoplasma) are outweighed by the Vibrio at Skokomish. However, qPCR assays, specifically with total 16S rRNA quantification and targeted Vibrio quantification are outside the bounds of this work, as they require additional method development to maximize efficiency and reliability.

Line 363 - minor grammar "(29)". **This has been removed, it was a citation artifact.**

February 17, 2022

Dr. Emily Kunselman
Scripps Institution of Oceanography
8750 Biological Grade
La Jolla, CA

Re: Spectrum01982-21R1 (**Variation in survival and gut microbiome composition of hatchery grown native oysters at various locations within the Puget Sound**)

Dear Dr. Emily Kunselman:

Link Not Available

Sincerely,

Konstantinos Kormas

Journals Department
Reviewer comments:

Reviewer #1 (Comments for the Author):

The authors have addressed all my comments satisfactorily.

Reviewer #2 (Comments for the Author):

The manuscript is well written and makes a significant contribution to the scientific community.

Reviewer #4 (Comments for the Author):

This manuscript investigated the bacterial community of the Olympia oyster in Puget Sound. Although the research has important implications, the methods and results are not novel. Other specific comments are as follows:

1. The experimental design, especially the sampling process, has great deficiencies. For example, the Case Inlet, which served as an important control, only 3 water samples were taken ($n = 2$ inside eelgrass beds and $n = 1$ outside eelgrass) in the study. At least 3 parallels are required to meet statistical requirements.
2. Supplemental Figure F showed that the sequencing depth of some samples were insufficient.
3. The association between survival rate and microbiome has not been sufficiently analyzed and explored.
4. Mycoplasma had been found in many molluscs, such as in abalones (some was more than 80% in the diseased individual). Although the similar findings were found in this study, related causes were not analyzed and discussed in the Discussion of the manuscript.
5. Similarly, the manuscript has somewhat overstated the conclusions and implications of the work by strongly focusing on role of microbiome. Attention on the status of oysters was insufficient. It is suggested to add the physiological indicators or some key immune index of the oysters, and also the growth rate (or detect changes in body weight or shell length), in addition to survival rate.

Reviewer #5 (Comments for the Author):

The manuscript entitled "Variation in survival and gut microbiome composition of hatchery grown native oysters at various locations within the Puget Sound" by Kunselman et al. investigates changes in microbial communities, characterized by 16S rRNA gene sequencing, associated with different tissues of the native Olympia oyster at various sites and habitats in the Puget Sound. The goal of this study is to determine the impact of field site and habitat on the oyster microbiome (mainly the gut) to assess ecosystem health and to inform possible restoration efforts of depleted oyster beds in the Puget Sound. Overall, the manuscript is well written, coherent, and well-presented and I enjoyed reading it. The experimental techniques and statistical analyses are appropriate. I do have some general comments and suggestions for improvement that would need to be addressed before the manuscript could be published.

General comments:

- 1/ Although restoration of keystone species' natural beds is crucial from an ecological, economical or societal standpoint, I am struggling a bit with the significance of this study and its contextualisation (Line 34 - 39). NGS technologies and the use of microbiome have clearly been instrumental in recent years to better understand interactions between host-pathogen-environment as well as characterizing the general health of habitats. However, the proposed justification to use metabarcoding to inform decision on site selection for restoration appears a bit "stretchy".
- 2/ Authors should be cautious in the way some conclusions are presented and should moderate some statements to avoid overreached findings. The experimental approach has limitations - which the authors have appropriately highlighted at the end of the discussion - that affect the significance of the results.
- 3/ One of the main limitations of the design, in my opinion, is the limited number of animals deployed per cage/site ($n = 10$). Estimating survivorship from such low number and consequently using survival as a proxy to evaluate the general health of the habitat and its adequacy for restoration purposes does not support one of the main outcomes. In addition, as noted by the authors, adding at least another time point for sampling would have greatly improved this study.
- 4/ The use of a reference taxon for determining ratio and therefore characterizing differential abundance between sites is a sound statistical approach. Have the authors considered other analyses such as Random Forest model or Pearson correlation to establish specific ASVs that can predict performances?

Staff Comments:

Preparing Revision Guidelines

Please return the manuscript within 60 days; if you cannot complete the modification within this time period, please contact me. If you do not wish to modify the manuscript and prefer to submit it to another journal, please notify me of your decision immediately so that the manuscript may be formally withdrawn from consideration by Microbiology Spectrum.

This manuscript investigated the bacterial community of the Olympia oyster in Puget Sound. Although the research has important implications, the methods and results are not novel. Other specific comments are as follows:

1. The experimental design, especially the sampling process, has great deficiencies. For example, the Case Inlet, which served as an important control, only 3 water samples were taken (n = 2 inside eelgrass beds and n = 1 outside eelgrass) in the study. At least 3 parallels are required to meet statistical requirements.
2. Supplemental Figure F showed that the sequencing depth of some samples were insufficient.
3. The association between survival rate and microbiome has not been sufficiently analyzed and explored.
4. *Mycoplasma* had been found in many molluscs, such as in abalones (some was more than 80% in the diseased individual). Although the similar findings were found in this study, related causes were not analyzed and discussed in the Discussion of the manuscript.
5. Similarly, the manuscript has somewhat overstated the conclusions and implications of the work by strongly focusing on role of microbiome. Attention on the status of oysters was insufficient. It is suggested to add the physiological indicators or some key immune index of the oysters, and also the growth rate (or detect changes in body weight or shell length), in addition to survival rate.

Response to Reviewers

Reviewer #4

This manuscript investigated the bacterial community of the Olympia oyster in Puget Sound. Although the research has important implications, the methods and results are not novel. Other specific comments are as follows:

1. The experimental design, especially the sampling process, has great deficiencies. For example, the Case Inlet, which served as an important control, only 3 water samples were taken (n = 2 inside eelgrass beds and n = 1 outside eelgrass) in the study. At least 3 parallels are required to meet statistical requirements.

Due to low sample number for seawater and sediment, statistics were not performed to compare seawater or sediment across sites or habitats. Instead, results were focused on the oyster tissues when comparing sites and habitats. Seawater and sediment samples were just used to see differences between sample types.

2. Supplemental Figure F showed that the sequencing depth of some samples were insufficient.

Response to a previous reviewers' comments on rarefaction:

“Sequencing depth may be considered low in comparison to other microbiome analyses. In this study, we especially had to account for the lower diversity exhibited by oyster guts. Some (not all) prior literature shows similar rarefaction for oysters/ oyster guts (Dubé, Ky, and Planes 2019; Trabal Fernandez et al 2014). We have added the rarefaction curves to the supplementary data to demonstrate that the diversity plateaus early, justifying the ability to rarefy at 1,920 for all sample types, and at 1,000 for gut samples only”

“An initial rarefaction depth of 1920 was selected because the rarefaction curve shows that Shannon alpha diversity plateaus for all sample types after ~2000 sequences. However, gut samples have much lower sequence counts than other samples, and in order to retain most of the gut samples, we must rarefy below 2000 sequences.”

In summary, many oyster tissue samples had low sequence counts. Therefore, if data were rarefied at a higher number of sequences, many of the oyster gut samples would be lost. This would be a problem because the manuscript is heavily focused on oyster gut samples. Although not all samples' richness has plateaued at the chosen rarefaction depth, their Shannon diversity does plateau early.

3. The association between survival rate and microbiome has not been sufficiently analyzed and explored.

A previous reviewer commented that “The experimental design does not allow to establish a relationship between particular taxa and survival”, and we agree with this statement, and have noted already in the manuscript that “the microbiomes of dying oysters could not be captured in this study, [but] the patterns between survival rate and bacterial differentials suggest a potential role of these bacteria in oyster mortality, which should be further tested” (lines 384-

387), and “microbiome data was only collected for one time point in the late summer... the time of mortality for any lost oysters was unknown” (lines 416-420).

To further clarify this limitation, we have added the following sentence to the text: “Furthermore, it was not possible to statistically test correlation between survival rate and microbiome because the time of mortality for oysters at each site was unknown and microbiome of dying oysters was not captured” (lines 422-424).

4. Mycoplasma had been found in many molluscs, such as in abalones (some was more than 80% in the diseased individual). Although the similar findings were found in this study, related causes were not analyzed and discussed in the Discussion of the manuscript.

We believe Mycoplasma has been discussed to a thorough extent in the discussion (lines 316-32, 342-343, 364-369, and 376). Specifically, we state that “that Mycoplasma are likely relying on the oyster to provide certain compounds” (lines 320-321) and “Mycoplasma is characterized as a core member of the oyster gut in this study and associated with higher survival. One study found that Mycoplasma actually increased in proportion in the gills of disturbed oysters (59), but as they are normally identified in the gut, this could be a sign of inappropriate translocation from the gut to more distal tissues, suggesting physiological disturbance. Therefore, the high prevalence of gut-associated Mycoplasma in our study is unlikely to be a sign of disturbance” (lines 364-369).

5. Similarly, the manuscript has somewhat overstated the conclusions and implications of the work by strongly focusing on role of microbiome. Attention on the status of oysters was insufficient. It is suggested to add the physiological indicators or some key immune index of the oysters, and also the growth rate (or detect changes in body weight or shell length), in addition to survival rate.

We have previously responded to a reviewer comment about oyster health:

Reviewer mentioned: “The manuscript could be improved if the authors could provide other metrics for oyster health, such as growth rate, condition or presence of biofouling.”

Our response: “Unfortunately, the length of the oysters could not be measured at the beginning of the field exposure, so we do not have growth data. Additionally, we do not have any weight measurements (of tissue or whole shell). We do have the lengths of the oysters at the end of the experiment. The lengths do vary significantly across the sites (single factor ANOVA, $p = 0.006$ for shell width, $p = 0.038$ for shell length), but we do not think this is worth reporting because we do not have the change in length, and therefore cannot make any assumptions about differences in growth across sites. We do not have qualitative metadata for biofouling, but all the “Cages were cleaned of biofouling organisms and debris every two weeks during the deployment” (lines 94-95) as a protective measure for the oysters. “

Additionally, we cannot report any more data than is already presented in the manuscript or metadata file because no other data was collected. The focus of our manuscript is the microbiome and the implications of the microbiome findings.

Reviewer #5

The manuscript entitled “Variation in survival and gut microbiome composition of hatchery grown native oysters at various locations within the Puget Sound” by Kunselman et al. investigates changes in microbial communities, characterized by 16S rRNA gene sequencing, associated with different tissues of the native Olympia oyster at various sites and habitats in the Puget Sound. The goal of this study is to determine the impact of field site and habitat on the oyster microbiome (mainly the gut) to assess ecosystem health and to inform possible restoration efforts of depleted oyster beds in the Puget Sound.

Overall, the manuscript is well written, coherent, and well-presented and I enjoyed reading it. The experimental techniques and statistical analyses are appropriate. I do have some general comments and suggestions for improvement that would need to be addressed before the manuscript could be published.

General comments:

1/ Although restoration of keystone species’ natural beds is crucial from an ecological, economical or societal standpoint, I am struggling a bit with the significance of this study and its contextualization (Line 34 – 39). NGS technologies and the use of microbiome have clearly been instrumental in recent years to better understand interactions between host-pathogen-environment as well as characterizing the general health of habitats. However, the proposed justification to use metabarcoding to inform decision on site selection for restoration appears a bit “stretchy”.

We have reworded the significance statement to highlight those microbial dynamics could provide additional information when assessing quality of potential restoration sites because we found variability in survival and variability in microbiome (lines 34-38).

2/ Authors should be cautious in the way some conclusions are presented and should moderate some statement to avoid overreached findings. The experimental approach has limitations - which the authors have appropriately highlighted at the end of the discussion – that affect the significance of the results.

In response to the 1st comment, we removed a sentence from the conclusion that stated, “This study shows that some areas of Puget Sound may be less amenable to Olympia oyster restoration than others, which could guide the direction of restoration efforts”. Otherwise, we believe our discussion of the results and thorough investigation of limitations make a strong effort to announce our findings while also considering all the factors that could impact those findings.

3/ One of the main limitations of the design, in my opinion, is the limited number of animals

deployed per cage/site (n = 10). Estimating survivorship from such low number and consequently using survival as a proxy to evaluate the general health of the habitat and its adequacy for restoration purposes does not support one of the main outcomes. In addition, as noted by the authors, adding at least another time point for sampling would have greatly improve this study.

We previously responded to a reviewer comment on sample size by explaining, “there were 31 oysters from eelgrass and 27 oysters from unvegetated habitat across all the sites” and “after finding no differences in the microbiome between eelgrass and bare habitat, we technically have 2 cages per site”.

Furthermore, we do not directly correlate survival to microbiome, and we point out that we just observed an association between survival and microbiome that could not be statistically proven because we do not have information on the time of oyster mortality or the microbiome of a dying oyster (lines 422-424). We also state in the abstract that “Further work is needed to identify the specific bacterial dynamics that are associated with oyster physiology and survival rates” (lines 29-31).

We are careful not to use terminology that claims statistical significance of microbiome in mortality. For example, we use the word “association” rather than “correlation” to explain our observation that certain taxa are more prevalent in the sites where more oysters died. We do not perform any statistical test to try and correlate microbial ratios with survival rates.

4/ The use of a reference taxon for determining ratio and therefore characterize differential abundance between site is a sound statistical approach. Have the authors considered other analyses such as Random Forest model or Pearson correlation to establish specific ASVs that can predict performances?

We previously ran ANCOM and Random Forest with Qiime 2 and find consistent results with the ASVs which were differentially abundant across sites (ANCOM) or were predictive of site (Random Forest).

ANCOM (W > 6,000):

(4) Verrucomicrobiales

(2) Vibrio

(2) Synechococcus_CC9902

Desulfocapsaceae

(2) Unknown Gammaproteobacteria

Sulfurovum

Sedimenticolaceae

Halioglobus

Sandaracinaceae

Desulfobacterales

Desulfuromonadia

Random Forest:

Important explanatory features...

(2) Verrucomicrobiales (Rubritaleaceae family)

(1) unknown Proteobacteria

(1) unknown and (1) known Gammaproteobacteria

The accuracy ratio for this Random Forest was 2.5, with the overall accuracy at 66.7%, and you can see the heatmap of predictive ability below:

Although Port Gamble is harder to differentiate from the other sites, there is still an ability to partially predict which site the oyster came from based on its gut microbiome.

April 17, 2022

Dr. Emily Kunselman
Scripps Institution of Oceanography
8750 Biological Grade
La Jolla, CA

Re: Spectrum01982-21R2 (**Variation in survival and gut microbiome composition of hatchery grown native oysters at various locations within the Puget Sound**)

Dear Dr. Emily Kunselman:

Your manuscript has been accepted, and I am forwarding it to the ASM Journals Department for publication. You will be notified when your proofs are ready to be viewed.

Sincerely,

Konstantinos Kormas
Editor, Microbiology Spectrum

Journals Department
Supplemental File: Accept